# Laparoscopic assisted percutaneous herniorrhapy in dogs using PIRS technique

Przemysław Prządka[1]*, Bartłomiej Liszka[1], Piotr Skrzypczak[1], Dominika Kubiak-Nowak[1], Wojciech Borawski[1], Łukasz Juźwiak[1], Zdzisław Kiełbowicz[1], Dariusz Patkowski[2]

1 Department of Surgery, Faculty of Veterinary Medicine, Wroclaw University of Environmental and Life Sciences, Wroclaw, Poland, 2 Department of Pediatric Surgery and Urology, Medical University of Wroclaw, Wroclaw, Poland

* przemyslaw.przadka@gmail.com, przemyslaw.przadka@upwr.edu.pl

**Data Availability Statement:** All relevant data are within the manuscript.

**Funding:** The author(s) received no specific funding for this work.

## Abstract

### Background

In the literature, no studies describe the use of laparoscopic techniques for inguinal hernia repair in canine clinical patients. The surgical technique—Percutaneous Internal Ring Suturing (PIRS) presented in this article is the world's first minimally invasive laparoscopic surgical technique for inguinal canal closure in dogs.

### Aim

The aim of the presented study is to evaluate the possibility of employment of the laparoscopic PIRS technique in clinical practice as well as to technically evaluate its usefulness in the treatment of the inguinal hernia in dogs. The presented study describes the procedure and the results of laparoscopic treatment of 11 inguinal hernias in nine dogs (two bilateral).

### Methods

The whole procedure is performed under camera control introduced through one umbilical trocar. The very closure of the inner inguinal ring is done with the injection needle by a small puncture over the inguinal canal with the help of which the suture material is inserted, and the canal of the operated inguinal hernia is closed. Each operated dog underwent a thorough clinical examination before surgery which was combined with ultrasound examination of the inguinal canals before, immediately after and three months after surgery.

### Results

There was no hernia recurrence in the study period in the operated dogs and all individuals returned to full mobility immediately after recovery from anesthesia.

### Conclusions

Due to the low invasiveness and simplicity of performance, the PIRS technique described in the study should be taken into account when choosing a treatment method for non-traumatic

**Competing interests:** The authors have declared that no competing interests exist.

inguinal hernia in dogs. At the same time, the possibility of using the described technique in other types of inguinal hernia in dogs requires further research.

## Introduction

In dogs, the congenital inguinal hernia is more common in males than in females. An acquired inguinal hernia is more common in bitches, which was confirmed by Waters [1] who demonstrated in his research that 73% of acquired inguinal hernia concerns females. Most often, this disease occurs in dogs of small and medium breeds [2]. The most commonly predisposed breeds of dogs are Besenji, Pekingese, Poodle, Basset hound, Cairn terrier, Chihuahua, Cocker spaniel, Dachshund, Pomeranian, Maltese and West highland terrier [3]. The unilateral inguinal hernia is more likely to be right-sided than left-sided. The contents of the hernial sac may include, among others: omentum, fat, uterus, small intestine, colon, bladder and spleen [4, 5]. Entrapment of these tissues and organs can lead to a life-threatening situation [4].

The standard surgical approach is based on inguinal canal or midline dissection that allows for simultaneous repair of the bilateral inguinal hernia [6–8]. The latter surgical approach often requires additional tissue dissection around the mammary gland, increasing the risk of postoperative complications. Every open surgery method is based on the dissection of the hernial sac and reposition of its contents into the abdominal cavity. The emptied hernia sac is then ligated as close as possible to the inner inguinal ring. After amputation of the distal part of the hernial sac, the outer inguinal ring is closed with sutures [7, 8]. The invasive character of such a surgical procedure may lead to an increased risk of complications such as: incisional infection, wound dehiscence, hematoma, seroma, excessive postoperative swelling, hernia recurrence, sepsis or peritonitis and death. [9].

Laparoscopic inguinal hernia repair is widely used in human medicine. The techniques used vary depending on the patient's age [10–13]. The use of endoscopic techniques in humans allows, among others, to reduce trauma of performed operation and to significantly accelerate the recovery and the return to normal physical activity.

Laparoscopic inguinal hernia repair in canine clinical patients has not been described yet. Presented literature reports suggest theoretical use of the three-trocar technique which is similar to human laparoscopy [14]. Yet, this involves the surgeon having advanced manual skills and an extensive experience, especially in endoscopic suturing [12]. Another option is to perform inguinal hernia surgery assisted by laparoscopy. However, this procedure requires using two trocars, as well as making an additional incision over the inguinal canal that allows its closure [14]. In the surgical treatment of inguinal hernia in dogs, the NOTES (Natural Orifice Translumenal Endoscopic Surgery) technique was experimentally used. In this technique, the surgical approach was obtained through the stomach. After two weeks of clinical observation and later post-mortem examination, the authors assessed the risk of postoperative infections and found the method to be safe [15].

One of the methods of laparoscopic treatment of the inguinal hernia used in pediatric surgery is a procedure described by Patkowski et al. [13]. The technique is called Percutaneous Internal Ring Suturing (PIRS). It is an endoscopic technique in which, under the control of a laparoscope, the inner inguinal ring is closed percutaneously with a non-absorbable suture [12, 13, 16, 17]. This technique is different from all other laparoscopic inguinal hernia repair techniques in humans, since an almost excellent cosmetic outcome is achieved through the use

of a single port from the umbilicus. Extra-corporeal suturing is easy with this technique, and it does not require much experience [18–20].

This paper, according to the author's knowledge, is the first study presenting the results of the inguinal hernia treatment in dogs using the PIRS technique. Up to present, it has been the only laparoscopic method used in veterinary medicine that allows to attain closure of the inguinal hernia canal with minimal tissue injury. The described method, unlike other laparoscopic methods, requires the use of only one port to provide laparoscopic access for optics. Additionally, in comparison with other laparoscopic procedures, as well as with classical surgery used in companion animals, the presented technique radically reduces the procedure trauma. The significant advantages of the PIRS method include, among others, the ability to accurately diagnose and repair both open inguinal canals with simultaneous surgical treatment without increasing the invasiveness of the procedure. What is more, the method does not require additional incisions over the inguinal canal. This allows for achieving a very good cosmetic effect. The excellent results of PIRS technique in the treatment of the inguinal hernia in children prompted the authors to examine the possibility of adapting the technique for the treatment of the inguinal hernia in dogs. Thus, the paper presents the surgical technique as well as the results and assessment of the effectiveness of the PIRS in the treatment of the inguinal hernia in dogs.

## Materials and methods

The presented PIRS procedures were performed on patients of the Department and Clinic of Surgery at Wrocław University of Environmental and Life Sciences after obtaining the written consent of the animal owners for the procedure. The studies were approved by the II Local Ethical Committee for Experiments on Animals in Wroclaw (Resolution no. 063/2019). The research was conducted between 2016 and 2019. All surgical procedures and additional tests were carried out by the same team of doctors with a very extensive experience in their field.

Surgical treatment was administered to nine dogs (6 females, 3 males) aged 3 months to 5.5 years (mean age 2.25 years) and body weight from 2.1kg to 6.5 kg, (mean weight 3.6kg). Eleven inguinal hernias (in two dogs on both sides) were repaired laparoscopically. The dogs underwent clinical and basic blood tests before being qualified for surgery. Additionally, ultrasound examination of operated inguinal canals was performed before and immediately after surgery, and also three months later to check the inguinal canals operated.

### Description of ultrasound examination

The abdominal ultrasound examination was performed using a microconvex probe with the frequency of 3–9 MHz and a linear probe with the frequency of 4–13 MHz (Esaote MyLab Class C) before and immediately after the laparoscopic procedure, as well as 3 months from the day of the procedure. To carry out the procedure, the animal was placed in the dorsal recumbency. The hair was removed with an electric hair clipper, and ultrasound gel was applied to the hairless skin.

### Anesthesia and pre- and postoperative management

A complete blood count and biochemistry panel were performed in all cases. After conducting the clinical examination and taking into consideration the results of current blood tests, there were no contraindications to general anesthesia. Patients were premedicated intramuscularly with a mixture of dexmedetomidine (Dexdomitor, Orion Pharma) at a dose of 5mcg/kg with methadone (Comfortan, Dechra) at a dose of 0.2 mg/kg. Then, general anesthesia was induced with propofol (Scanofol, ScanVet) dosed according to the effect (usually approximately 1 mg/

kg). The patient was intubated subsequently. After tracheotubus was sealed and connected to the Datex Ohmeda S5 inhalation anesthesia apparatus, anesthesia was maintained with isoflurane (Isovet, Piramal Healthcare). Intraoperative anesthesia was obtained by continuous infusion of fentanyl (Fentadon, Dechra) at 0.2 mcg/kg/min after previous bolus administration at 2.5 mcg/kg. Treatment of postoperative pain included administration of buprenorphine (Bupaq Multidose, Orion Pharma) at a dose of 20mcg/kg every 8h for the next 3 days, and meloxicam (Metacam, Boehringer Ingelheim), also for 3 days—initially at a dose of 0.2mg/kg, then 0.1mg/kg. In addition, on the first day, patients were given metamizole (Pyralgivet, Vet-Agro) at a dose of 25–50mg/kg every 8 hours.

## Operative technique

PIRS treatments in dogs were performed using original method presented by Patkowski et al. [13]. All endoscopic equipment used for laparoscopic procedure with 5mm 30˚ scope was manufactured by Karl Storz SE & Co. KG (Tuttlingen, Germany). The patient was in the dorsal recumbency. Pneumoperitoneum ($CO_2$) was established with an open technique by introducing a 5-mm reusable trocar through a longitudinal incision at the umbilicus. Insufflation pressure was between 8–10 mm Hg, based on the patient's size. The whole peritoneal cavity was inspected. Any hernia was reduced manually or with the aid of the telescope tip. All the needle movements were performed from outside the body cavity under direct camera control. To choose the location for the needle puncture, the position of the internal inguinal ring was assessed by pressing the inguinal region from the outside with the tip of Pean forceps (Fig 1A and 1B). Under laparoscopic-guided vision the 18-gauge injection needle with nonabsorbable 2–0 polyfilament (Ti-Cron™, Covidien) thread inside the barrel of the needle was introduced through the abdominal wall entering the abdominal cavity at the middle upper outline of internal inguinal ring. With the movements of the tip of the needle, the thread was passed under the peritoneum, over half of the internal ring including a part of the ligament and adjacent tissue (Fig 1C and 1D). The thread was pushed through the barrel of the needle into the abdominal cavity making a loop (Fig 1E). The needle was pulled out, leaving the loop of the thread inside the abdomen (Figs 1F and 2A). From outside the dog's body, one of the thread ends was introduced again into the barrel of the needle and the needle passed through the same skin puncture point to surround the other half of the internal ring with part of the round ligament (Fig 2B and 2C). In order to prevent the vas deferens and testicular vessels from injury, a small space was left above these structures. The end of the thread went through the barrel of the needle into the thread loop, and the needle was withdrawn (Fig 2D and 2E). Next, the thread loop was pulled out of the abdomen with the thread end caught by the loop (Figs 2F and 3A). In this way, the thread was placed around the inguinal ring under the peritoneum and both ends exited the skin through the same puncture point (Fig 4). The knot was tied to close the internal ring and was placed under the skin (Fig 3B). The incision in the umbilicus was closed in layers with single absorbable sutures (Fig 3C).

## Mathematical calculations

In the presented article all calculations were done using Microsoft® Office Excel.

## Results

In the clinical examination, palpation revealed uni- or bilateral deformities at the level of the inguinal canal. In all of the cases, it was possible to reduce the hernia content into the abdominal cavity with gentle massage of the hernia sac.

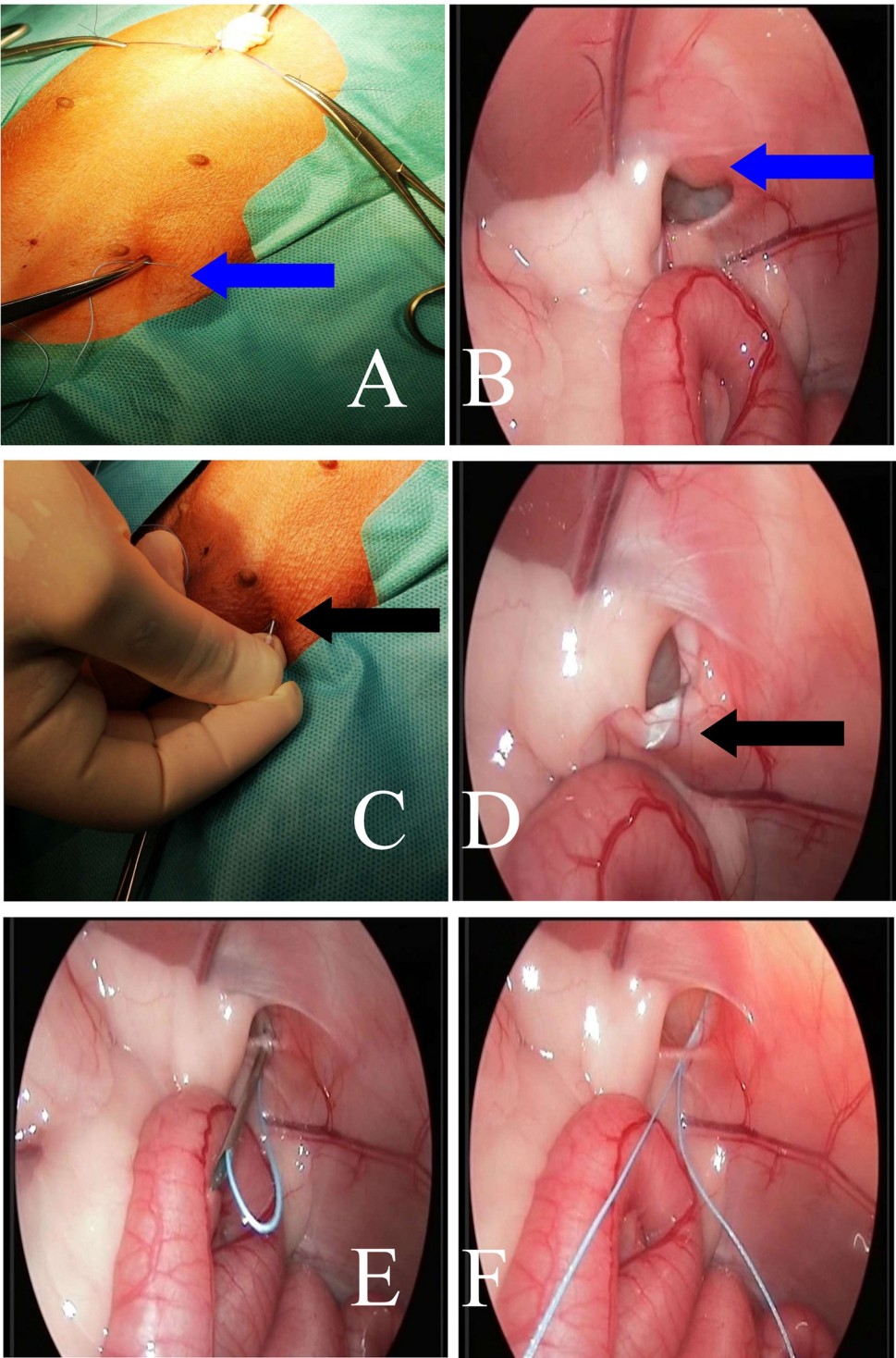

**Fig 1. Intraoperative image.** A, B—Determination of the place of insertion of the needle and the insertion of the suture over the operated inguinal canal (blue arrows) A—external image, B—laparoscopic image of the abdominal cavity. C, D—Insertion of the needle into the previously designated place and its passage through the outer half of the inner inguinal canal ring (black arrows), C—external image, D—laparoscopic image of the abdominal cavity. E—Introduction of the suture through the needle in order to create a loop—laparoscopic image of the abdominal cavity. F—View of the loop of polyester suture after removing the injection needle—laparoscopic image of the abdominal cavity.

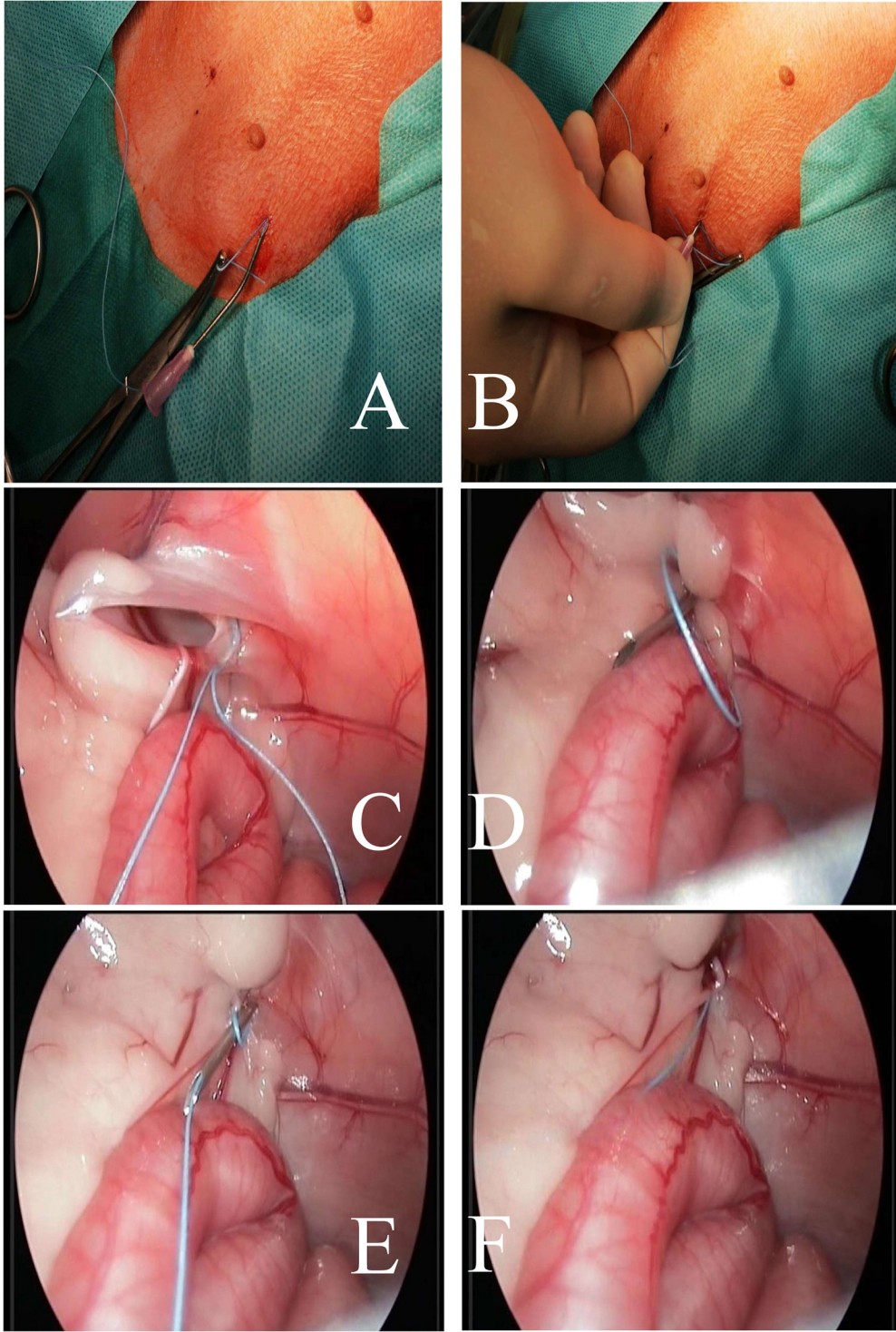

**Fig 2. Intraoperative image.** A—External image of the removed needle after the loop has been made. B—External image of the needle insertion in the same place where the needle was previously inserted. Before the insertion, the part of suture at the tip of the needle was cut off from the part of the suture forming the loop in the abdominal cavity. The external portions of the loop were secured with hemostatic forceps. C, D, E, F—Stages of passing the needle through the medial part of the inner ring of the inguinal canal and introducing the polyester suture placed in the needle through the previously formed loop.

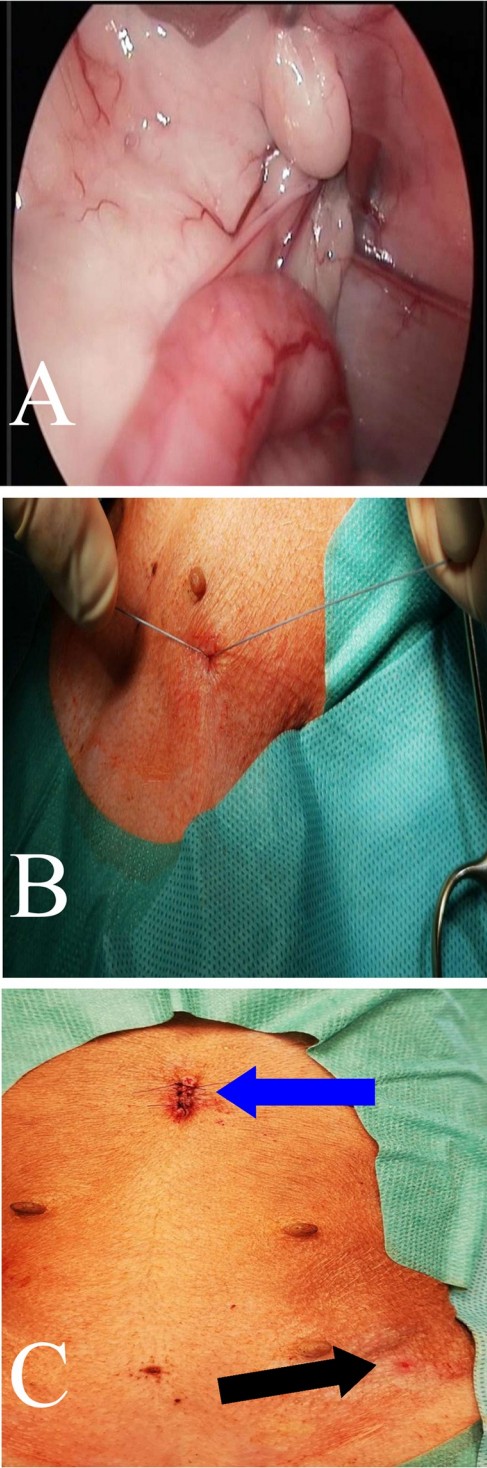

**Fig 3. Intraoperative image.** A—Laparoscopic image of the closed inguinal canal. B—Knotting a polyester suture closing the inner ring of the inguinal canal. C—Image of postoperative wounds. The blue arrow indicates the wound after insertion of the optical trocar of the laparoscope. The black arrow indicates the site of insertion of the needle above the inguinal canal.

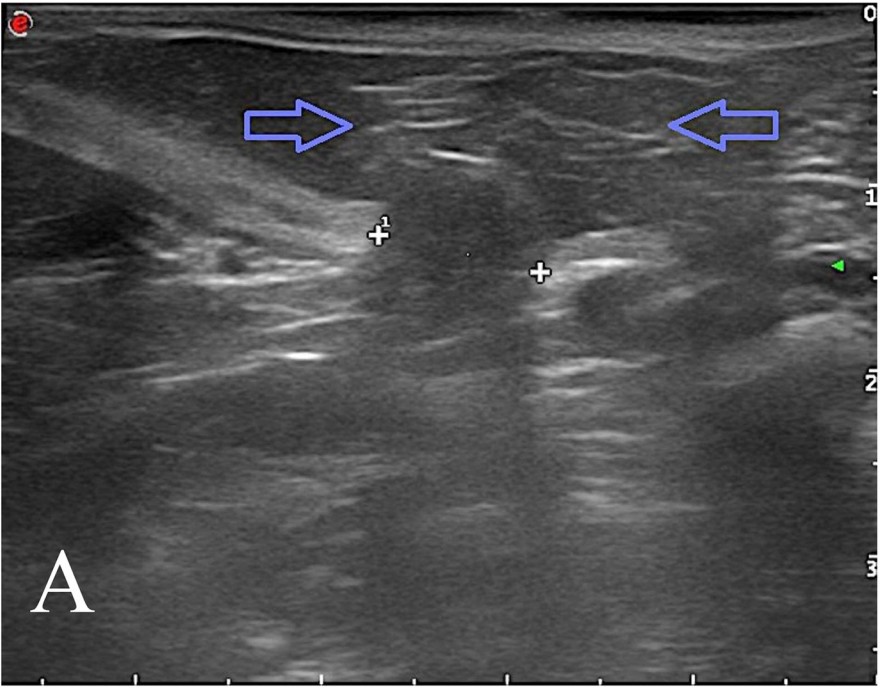

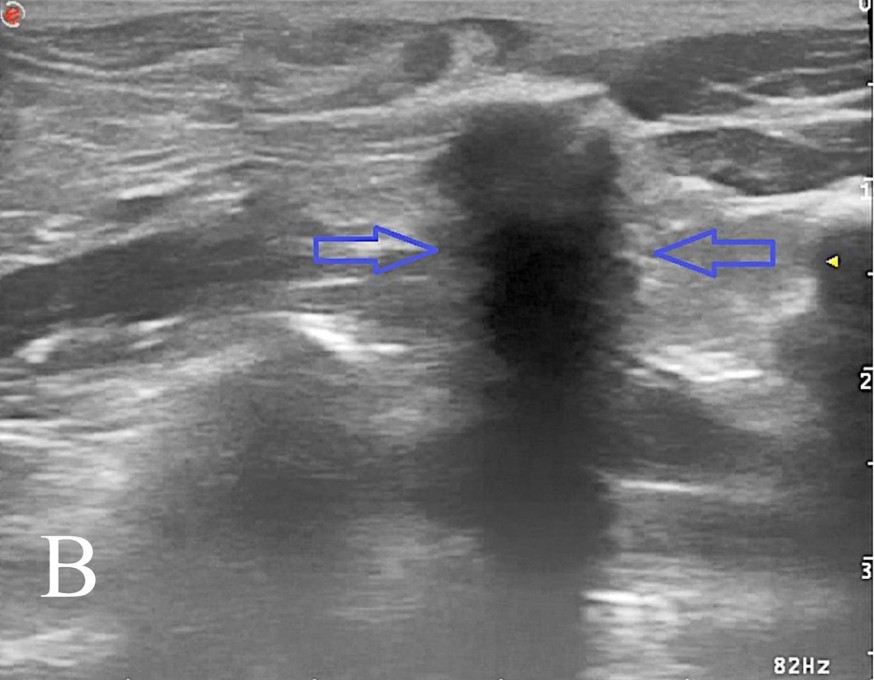

**Fig 4. Ultrasound image of inguinal hernia in a male dog.** A—Before laparoscopic surgery: the hernial sac contained peritoneal fat (blue arrows); the width of the hernia gates was marked between the cursors. B– 3 months after laparoscopic surgery: hyperechoic, clear acoustic shadow (blue arrows) surgical suture is marked, hernia gate is closed. No evidence of inguinal hernia recurrence was noted.

**Table 1. Listing and additional information about operated dogs.**

| Breed | Age (mo) | Sex | Weight (kg) | Side of the hernia | Anesthesia time (minutes) | Time of surgery (minutes) |
|---|---|---|---|---|---|---|
| Yorkshire terrier | 24 | female | 2,1 | left | 35 | 25 |
| Yorkshire terrier | 12 | female | 2,7 | left | 30 | 23 |
| Mix | 66 | female | 6,5 | left | 42 | 30 |
| Shih tzu | 3 | male | 2,1 | left/ right | 35 | 25 |
| Mix | 36 | male | 4,4 | left/ right | 41 | 32 |
| Shith tzu | 30 | female | 3,2 | right | 31 | 22 |
| Yorkshire terrier | 18 | male | 2,9 | left | 30 | 20 |
| Mix | 24 | female | 5,1 | left | 32 | 23 |
| Maltese | 36 | female | 3,4 | left | 33 | 24 |

All laparoscopic procedures were performed without intraoperative complications that required conversion to open surgery. The mean duration of anesthesia was 29 ± 4.6 minutes and the mean operating time was 19.5 ± 3.4 minutes. In bilateral hernias, the mean duration of anesthesia was 38 ± 4.2 minutes and the mean operating time was 28.5 ± 4.9 minutes. The duration of the operation was defined as the time between the beginning of preoperative washing with preparation of the surgical field (induction of pneumoperitoneum) and the suturing of the wound in the site where an optical trocar was inserted. Table 1 presents the details of the operated animals such as: age, gender, type of the inguinal hernia (right, left, bilateral), the duration of anesthesia and the surgery.

Based on abdominal ultrasound findings, it was revealed there was breakdown in the continuity of the abdominal wall in the groin area with well seen hernia gates. The contents of the hernial sac were found to be peritoneal fat and dislocation of abdominal organs was not observed (Fig 4A). In a control ultrasound performed immediately after and 3 months after the laparoscopic procedure, the hernia gates were closed and the acoustic shadow from the suture material was visible (Fig 4B).

No recurrent hernia was found in any of the operated dogs. This was confirmed by both clinical and ultrasound examination 3 months after surgery. The operated dogs returned to full physical activity immediately after recovery from anesthesia. In three bitches, a small volume of medical carbon dioxide trapped in the hernial sac was observed immediately after the procedure after closing the inner ring of the inguinal canal. The visible deformation disappeared completely within a few hours. At the same time, no clinical signs of gonadal circulation disturbance were observed in the operated males. None of the operated dogs had swelling or other disturbing symptoms resulting from the procedure.

The wounds in the site where the optical trocar was inserted healed by the first intention leaving a barely visible 0.5cm long linear scar. The places where the injection needle was inserted, through which the suture material closing the inner inguinal ring was introduced and tied, were invisible. At the same time, no abnormalities were found in the palpation (thickening, fistula, palpable suture knots).

## Discussion

Laparoscopic inguinal hernia surgery is now standard practice in many human surgical centers. In veterinary medicine literature, despite the dynamic development of endoscopic surgery, only a few references are available on the possibility of laparoscopic inguinal hernia repair in dogs. All papers available to authors present laparoscopic inguinal canal closure using three or two-trocar surgical access. These treatments are modeled on human medicine

[14]. They usually require intra-abdominal sewing, which is considered one of the most difficult technical parts in laparoscopic procedures. The disadvantage of this method, apart from the fact it requires a highly-skilled endoscopic sewing surgeon to perform it, is that the operating procedure itself is time-consuming. In addition, manipulation of instruments inside the abdominal cavity increases the risk of damage to blood vessels, viscera and nerves [13]. Monnet and Fransson [14] suggest that, theoretically, the three-trocar laparoscopic method of closing the inguinal canal can be employed in dogs. The method was adapted from human medicine and it consists in stitching the hernia gates inside the abdomen. Yet, the authors point out that laparoscopic repair of the inguinal hernia has not yet been reported in a series of dogs or cats [14]. Sousa et al. [21] present an experimental study into the application of various prostheses in the laparoscopic treatment of inguinal hernia in dogs. Yet, the authors do not describe the endoscopic technique used in the surgery. Among the methods of surgical treatment of the inguinal hernia with laparoscopy, the two-trocar technique was described. In this method, after the hernial sac had been reduced into the abdominal cavity, the hernial gates were closed under the control of a laparoscope with a suture around the hernial sac. The suture was introduced through a small incision in the skin, which was made over the operated inguinal canal [14]. In experimental studies in dogs, Ger at al. [22] presented the possibility of using staplers in laparoscopic closure of the inguinal hernia sac. In their procedures, they used two trocars. After laparoscopic assessment of the inguinal canals, the working canal was always on the opposite side to the operated inguinal hernia. In case of bilateral hernias, the procedure would require the use of three trocars. Sherwinter et al. [15] presented the results of experimental treatment of the inguinal hernia in five dogs using the NOTES (Natural Orifice Translumenal Endoscopic Surgery) technique. In this technique, the approach was obtained via the stomach wall. Open hernia canals were closed with acellular dermal implant, which was deployed across the entire myopectineal orifice and draped over the cord structures. The implant was secured against dislocation by BioGlue introduced transdermally with an injection needle. Short-term (14 days) clinical observations, as well as post-mortem and microbiological examinations of the operated dogs, revealed no general and local infections, therefore, confirming the safety of NOTES technique in the treatment of the inguinal hernia. [15].

The presented technique requires an introduction of only one umbilical trocar for laparoscopic scope and closure of the inner inguinal canal ring is done percutaneously using an easily available 18G injection needle. This allows to avoid using more trocars and any tools insertion into the abdominal cavity. Therefore, the risk of complications in the form of damage to internal organs is significantly reduced, especially since the procedures were performed on small patients (average weight 3.6 kg). The uncomplicated laparoscopic PIRS procedure does not require the surgeon to have advanced skills of laparoscopic techniques, which can be important in the case of laparoscopic veterinary surgery. This applies especially to surgeons with only basic endoscopic surgery skills. Apart from the necessity of owning a laparoscope, PIRS treatments are cheap as there is no use of specialized surgical instruments such as staplers. In a case of difficult internal ring visualisation it is always possible to introduce another trocar with additional instrument, however no patient in presented study required.

Another important aspect of the PIRS technique, which surpasses open surgery techniques for inguinal hernia repair, is the ability to evaluate and simultaneously repair a bilateral inguinal hernia or an incidentally recognized open inguinal canal on the opposite side. This applies especially to patients in which one of the bilateral hernias is small and often undiagnosed in a clinical examination. Nonetheless, the inguinal canal is open in these patients [7, 23]. In human medicine, it is assumed that there is about 10.2% risk of a hernia on the opposite side and the risk increases to over 19% if a primary hernia is left-sided [24]. The classic surgical technique in humans does not allow such an assessment. In the classic surgical treatment of

unilateral hernias in dogs, to detect the possible presence of a clinically undiagnosed hernia on the opposite side, it is necessary to make incisions over each of the inguinal canals or perform a median incision with partial dissection of the mammary gland laterally on both sides of the incision line to reach both inguinal canals. This significantly increases the trauma, the duration of the procedure and the risk of postoperative complications described below in the discussion. In situations without intraoperative assessment of the (clinically healthy) opposite inguinal canal, it may be necessary to carry out another procedure after the clinical manifestations of an undiagnosed and untreated inguinal hernia have been revealed at the time of the first procedure. Among the cases included in the presented study, two dogs with bilateral hernias diagnosed before surgery were operated. At the same time, in the remaining cases, the other inguinal canal was always evaluated to detect any possibly undiagnosed hernia that required operation during the same procedure. The possibility of closing the open inguinal canal without the clinical presentation of an inguinal hernia should be considered as prevention. Furthermore, an opportunity to repair inguinal hernia in both inguinal regions during one endoscopic procedure proves that the described method is significantly superior to other techniques of the classic surgery that have been used in the treatment of inguinal hernias so far. This is also confirmed by Ger at al. [22] who list an opportunity for viewing the intra-abdominal viscera and the ability to diagnose and treat bilateral hernias, common in pediatric patients, without further invasive surgery among important advantages of endoscopic surgery.

Complications in dogs treated surgically for an inguinal hernia include incisional infection, wound dehiscence, hematoma, seroma, excessive postoperative swelling, hernia recurrence, sepsis or peritonitis and death [9]. These complications often result from the extent of the procedure itself and occur after an open surgery [25, 26]. Waters et al. [1] conducted a retrospective study from several veterinary clinics and found that postoperative complications after open inguinal surgery in dogs accounted for 17% of all postoperative complications. Yet, the authors emphasized that, in their opinion, the calculation was made on the basis of a small number of clinical cases [1]. The trauma resulting from the PIRS method is limited to a small incision in the skin (one trocar) and two injection needle pricks. This significantly reduces the risk of postoperative complications. The authors of the presented study did not identify any of the above-mentioned postoperative complications. The obtained results are comparable with those reported by other authors who experimentally used endoscopic surgery to close inguinal canals in dogs [15, 21]. However, it should be taken into account that the groups of animals involved in the assessment of minimally invasive methods were small.

In all of 11 inguinal hernia repairs in nine dogs operated with the PIRS technique, no recurrences were reported. This was confirmed in both clinical and ultrasound examination three months after surgery. This is also confirmed by the feedback from the owners of the operated dogs six months after surgery. The authors assumed that the repeated ultrasound examination three months after the surgery would allow for an objective evaluation of possible late postoperative complications, such as hernia recurrence or a reaction to the suture material. In human medicine, a large number of patients have demonstrated a great clinical usefulness of ultrasound examination three months after laparoscopic inguinal hernia repair. The value of this examination was particularly significant in patients with clinical changes in the operated inguinal canal area which were present at the time [27]. According to reports from human medicine, the frequency of recurrence after laparoscopic inguinal hernia repair is slightly higher compared to open surgery. However, these differences are statistically insignificant [28, 29]. The highest percentage of postoperative relapses in children was recorded in boys over 1.5 years of age, yet, this was not observed in girls and boys under 3 months of age [13]. The authors of the cited study perceive the lack of complications in the youngest children as the result of the body's natural biological ability to close the inguinal canal after its narrowing with

PIRS technique [13]. The oldest and, at the same time, the heaviest patient in the study operated with the PIRS technique was a 3-year-old female weighing 6.5 kg. The lack of older, and above all, heavier dogs in the studied group of animals prevents the authors from explicitly assessing the PIRS technique in groups of large and Molossian dogs. The lack of circulatory disturbance in the gonads and the lack of compression of the vas deferens in operated males resulted from the fact that above the mentioned structures, the suture was laid leaving 1-2mm of space. Other authors also emphasize the importance of leaving adequate space during the inguinal hernia repair for the vas deferens and the testicular vessels [30]. In females, where there is no need to leave a small space for blood vessels and vas deferens running to the gonads, the procedure is much easier. Another potential complication reported in children operated with PIRS procedures is the accidental puncture of the iliac vessels, which can be controlled by momentary pressure from the outside under the control of an endoscopic camera. The bleeding is usually local and does not affect the further course of the procedure or the need for conversion. The possibility of damage to the iliac vessels in humans is related to the proximity of the inguinal canals and the abovementioned blood vessels. Such complications were not observed in the operated dogs. This may results from, among other reasons, a different anatomy and mutual arrangement of the described structures.

According to the authors, the undoubted advantage of the PIRS method is the short duration of the entire procedure. The intervention lasts less than 30 minutes for unilateral hernias. It is impossible to objectively compare the operative time to other methods hindered by lack of information in the literature regarding completely laparoscopic techniques [21, 22], assisted by laparoscopy [14] or open surgeries [31]. Sherwinter et al. [15] performed unilateral inguinal hernia repairs with the NOTES technique needed an average of 89 minutes (range 45–150). Yet, the shortest NOTES procedure was longer than the PIRS procedure, even for dogs with bilateral hernias. The situation is similar when the average time of the unilateral inguinal hernia repair with the PIRS technique (19.5 minutes on average) is compared to experimentally performed open surgeries in dogs. This concerns the duration of surgeries using own tissue hernioplasty (average 27 minutes), as well as those in which synthetic hernia mesh was used (average 33 minutes) [31].

The rarity of inguinal hernia in males is confirmed by many authors [1, 32]. However, the percentage of males with inguinal hernias in the general population of dogs affected by this condition is variable and, depending on the source of information, it usually ranges from 11% [32] to 37% [1]. It should be noted that despite the authors' agreement regarding the lower incidence of the inguinal hernia in male dogs, the percentage of incidence is correlated with the number of animals presented in the paper. The best example of this is the present work. In the study, more than a third of the dogs were males (three dogs), among which two-thirds had bilateral inguinal hernias (two dogs). Yet, in a group of 74 dogs diagnosed with inguinal hernia, Strande [32] found bilateral inguinal hernias only in nine males. Nonetheless, the most common in dogs are one-sided inguinal hernias [1], which in the presented work accounted for almost 80%. As many as 85% of them were left hernia cases. The main limitations of the PIRS technique are generally accepted contraindications to endoscopic surgery [33]. This refers to, for example, patients whose poor general condition is a contraindication to general anesthesia and the procedure can be performed under premedication and local epidural anesthesia. However, in patients without contraindications for laparoscopic surgery, one of the major advantages of the PIRS technique is its minimally invasive nature. This technique allows for reduction of postoperative pain and speedy return to normal functioning. Additionally, in cases of possible postoperative recurrence, the surgeon has a choice as to whether to perform reoperation with the same surgical method (PIRS), or to choose another surgical technique. This has been confirmed in human medicine [13].

The price of the procedure is also a limitation for laparoscopic treatment of inguinal hernia in companion animals. It is usually accepted that minimally invasive procedures are more expensive compared to open surgery methods. However, this results mainly from the fact that the purchase of expensive devices and instruments is necessary to perform endoscopic procedures. Yet, as shown in comparative studies of the costs of various laparoscopic and open surgery procedures, the price of laparoscopic cholecystectomy is lower than the price of open surgery procedures [34]. Nonetheless, the very popularity of the procedure should be a result of its advantages and disadvantages. This concerns mainly the invasiveness of the surgical method, possible complications, the time of hospitalization, and the time needed for the patient to return to the normal functioning after the surgery. The price of the procedure as the factor influencing the popularity of the method should be considered in the next order. Interesting results of retrospective studies conducted in human medicine show that the cost of a total laparoscopic treatment (cost of the surgery and hospitalization) is lower compared to the analogous open surgery procedures [35]. In case of veterinary clinics already equipped with a basic laparoscopic equipment, the price of performing inguinal hernia repair with the PIRS technique should not differ significantly from the price of the same procedure performed with an open surgery approach.

Another interesting issue is the limitation of using the PIRS technique in the treatment of post-traumatic inguinal hernias and/or non-reducible hernias. According to the authors, further research is needed into the possibility of using the PIRS technique in the treatment of post-traumatic inguinal hernias. In such cases, the present adhesions and the extent of hernia gates would probably be the factors that make the procedure difficult. It is likely that endoscopic treatment of these types of inguinal hernias will require the introduction of more than one trocar in order to allow manipulation inside the abdominal cavity. Closing hernia gates after successful reduction of the contents of the hernia can be done using the PIRS method. However, in these instances, more than one suture may be required, as it is the case in the non-traumatic and reducible hernias described by the authors. Such a repeated suturing with the "PIRS method" has been demonstrated in endosurgical treatment of the diaphragmatic hernia in children [36].

The presence of a hair cover in dogs makes the obtained cosmetic effect in the PIRS technique less important. On the other hand, smaller trauma, as well as the possibility of a quick return to normal functioning, is comparable in both humans and animals. All operated dogs returned to normal activity immediately after recovery from anesthesia. This is probably due to a minor trauma of the abdominal wall (0.5 incision for the optical trocar) and a minor trauma at the site where the inner inguinal ring is closed—namely, an injection needle prick, and tying the suture introduced by the needle. In addition, the short duration of the procedure has a positive effect on the well-being of the operated dogs.

## Conclusions

Based on the results of laparoscopic inguinal hernia repair using the PIRS technique, the authors conclude that this technique should be considered as one of the treatment options for inguinal hernias in dogs of small and medium breeds. At the same time, according to the authors, further clinical research in larger numbers of dogs of small and large breeds is needed. Such studies should also include cases of traumatic hernias, such as non-reducible non-obstructed hernias. Yet, some modification of the surgical method described in the presented study may be required in these cases.

## Author Contributions

**Conceptualization:** Przemysław Prządka.

**Data curation:** Przemysław Prządka, Bartłomiej Liszka, Piotr Skrzypczak, Dominika Kubiak-Nowak.

**Formal analysis:** Łukasz Juźwiak, Dariusz Patkowski.

**Investigation:** Piotr Skrzypczak, Wojciech Borawski.

**Methodology:** Przemysław Prządka, Dariusz Patkowski.

**Project administration:** Wojciech Borawski, Zdzisław Kiełbowicz.

**Supervision:** Łukasz Juźwiak.

**Visualization:** Przemysław Prządka, Dominika Kubiak-Nowak, Wojciech Borawski.

**Writing – original draft:** Przemysław Prządka, Bartłomiej Liszka.

**Writing – review & editing:** Zdzisław Kiełbowicz, Dariusz Patkowski.

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
