## [Decision Letter · Decision Letter 0]

19 Mar 2020

PONE-D-19-32639

Percutaneous Internal Ring Suturing - minimally invasive technique for inguinal hernia repair in dogs.

PLOS ONE

Dear Dr Przadka,

Thank you for submitting your manuscript to PLOS ONE. After careful consideration, we feel that it has merit but does not fully meet PLOS ONE’s publication criteria as it currently stands. Therefore, we invite you to submit a revised version of the manuscript that addresses the points raised during the review process.

I think that the topic it is very interesting in the minimally invasive surgery companion animals field. Some of the reviewers decided for reject of manuscript, and others for acceptance with revisions. I agree with some of the reviewers when they say that there are major flaws in the manuscript that should be revised in order to meet the journal’s requirements. The manuscript must be revised, with more attention in the English correction of some words, removing some unclear or complicated sentences. Also, special attention should be taken in order to improve Discussion and Conclusions items. So, in view of the criticisms of the reviewer(s) that you may found at the bottom of this letter, I decided that you may revise your manuscript and re-submitted it after major revisions.

We would appreciate receiving your revised manuscript by May 03 2020 11:59PM. To enhance the reproducibility of your results, we recommend that if applicable you deposit your laboratory protocols in protocols.io, where a protocol can be assigned its own identifier (DOI) such that it can be cited independently in the future. For instructions see: http://journals.plos.org/plosone/s/submission-guidelines#loc-laboratory-protocols

We look forward to receiving your revised manuscript.

Kind regards,

L. Miguel Carreira, PhD, MSc,DTO,Ps-Grd,DMD,DVM

Academic Editor

PLOS ONE

2. In your Methods, please state where the participants were recruited for your study.

3. ** Please include your tables as part of your main manuscript and remove the individual files **. Please note that supplementary tables (should remain/ be uploaded) as separate "supporting information" files.

Reviewers' comments:

Reviewer's Responses to Questions

**Comments to the Author**

1. Is the manuscript technically sound, and do the data support the conclusions?

Reviewer #1: Yes

Reviewer #2: Yes

Reviewer #3: Partly

Reviewer #4: Partly

Reviewer #5: Yes

2. Has the statistical analysis been performed appropriately and rigorously? 

Reviewer #1: Yes

Reviewer #2: N/A

Reviewer #3: No

Reviewer #4: No

Reviewer #5: N/A

3. Have the authors made all data underlying the findings in their manuscript fully available?

Reviewer #1: Yes

Reviewer #2: Yes

Reviewer #3: Yes

Reviewer #4: Yes

Reviewer #5: Yes

4. Is the manuscript presented in an intelligible fashion and written in standard English?

Reviewer #1: Yes

Reviewer #2: Yes

Reviewer #3: No

Reviewer #4: Yes

Reviewer #5: Yes

5. Review Comments to the Author

Reviewer #1: I read carefully this case series study it is interesting method with laparoscopic assisted herniorrhaphy in dog I think it could be useful and interesting for small animal practitioner and clinicians so I recommend strongly for publication.

I recommend to change the title as "laparoscopic assisted inguinal herniorrhaphy through the percutaneous insertion of injection needle"

Reviewer #2: Dear auther,

Based on reading of the presented study, the article structure is good and the results of origininal research in the veterinary practice and not puplished. The study idea or hypothesis not original as it was introduced in pedeatric surgery for the same point even with the similar topic:

“Percutaneous internal ring suturing: a simple minimally invasive technique for inguinal hernia repair in children”

- I sugesst to modify your reseach topic.

- I sugesst that the Experiments, statistics, needs more detail using specific statstical programs and tests.

- Did 3 months follow up are enough for juding?

- Is there more clearer ultrasound picture?

- In line 187 & 188, is the ultrasound description of the inguinal hernia “disruption in the continuity of the abdominal wall”. Please make the description revealant with pictures comments.

- The Conclusions are presented in an appropriate fashion and are supported by the recorded data. The article is presented in an intelligible fashion and is written in standard English. The research meets all applicable standards for the ethics of experimentation and research integrit.The article adheres to appropriate reporting guidelines and community standards for data availability

Reviewer #3: Dear authors this topic is very interesting, and the study is well done but unfortunately some major issues must be revised, and some sentences should be edited to meet the journal’s requirements. please find attached the comments.

Comment 1) I would suggest more attention in the English correction of some words. The authors must check their manuscripts by an English language native speaker before their submission.

Some sentences are unclear or complicated. For example:

Line 40-42, line 55, line 58-60, line 68-72, line 75-76, line 88-89, line 111, line 182-185, line 200-202, line 205-206, line 208-211, line 218-223, line 230-235, line 265-266

Some sentences need to be revised:

Line 34. In the literature there are no studies describing … Please reverse it as: In the literature, no studies are describing …

Line 42. Please add a comma after the word inserted

Line 44. It is recommended that the numbers below 10 should be spelled out please revise the sentence in 9 dogs as: in nine dogs.

Line 52. The noun congenital inguinal hernia is missing a determiner before it. Please revise it or change the sentence. For example: In dogs, the congenital inguinal hernia is more common … or Congenital inguinal hernias are more common (Plural)....

Line 53. The noun Acquired inguinal hernia is missing a determiner before it. Please revise it as: An acquired inguinal hernia is more common... or Acquired inguinal hernias are more common ...

Line 58. Please revise the sentence One-sided inguinal hernia is… as: unilateral inguinal hernias are... or the unilateral inguinal hernia is…

Line 62. Please revise the sentence repair of bilateral inguinal hernia as: repair of bilateral inguinal hernias or repair of the bilateral inguinal hernia (The noun is missing a determiner before it)

Line 64. The noun dissection is missing a determiner before it, please change it as: based on the dissection of …

Line 74 and 78. The noun inguinal hernia is missing a determiner before it, please change it as: inguinal hernias or the inguinal hernia

Line 82. Please add an article before the word author’s knowledge: according to the author's knowledge…

Line 94. The noun inguinal hernia is missing a determiner before it, please change it as: inguinal hernias or the inguinal hernia.

Line 99. The first letter of a sentence should be in upper case if multiple sentences are part of the same parentheses. (Resolution no. …)

Line 111. The sentence is correct, but the wording is not optimal please edit this sentence

Line 133. Please revise the word supine position with dorsal recumbency

Line 135. Please add a dot after the word size

Line 137. Please remove the word cavity or revise it as: outside of the abdominal cavity

Line 143. Please add a comma after the word needle

Line 152. Please add a comma after the word loop

Line 153. Please add a comma after the word way.

Line 156. The word umbilicus is missing a determiner, please add.

Line 160. The noun frequency is missing a determiner before it, please add.

Line 162-163. Please revise the sentence the animal was placed on its back in a

positioning trough, as: the animal was placed on the dorsal recumbency.

Line 163. Replace machine with clipper

Line 173. Please add an article before the word optical: where an optical …

Line 182. Please add an article before the word first: by the first intention …

Line 187. Please add an article before the word disruption

Line 189. Please add an article before the word dislocation

Line 198. Please revise the word trocar to the plural form

Line 208. Please add an article before the word introduction. For example: … requires an introduction of …

Line 215. The word doctor is correct, but it would be much better if you replace it with veterinarians or surgeons

Line 220. Please add an article before the word bilateral: a bilateral …

Line 224. Please add an article before the word primary: if a primary

Line 225. Please add an article before the word classical (two times)

Line 226. Please revise the unilateral hernia to the plural form: unilateral hernias …

Line 226. Please delete the word in order. The phrase in order to may be wordy. Consider changing the wording with to.

Line 234. Please revise the word hernia to the pleural form: hernias

Line 236. Please delete the word in order

Line 238. Please revise the sentence presence of inguinal hernia to presentation of an inguinal hernia (The word presence does not seem to fit in this context. Consider replacing it with presentation)

Line 238. Remove the word a please (The indefinite article, a, may be redundant when used with the uncountable noun prevention in your sentence)

Line 241. Please revise the word hernia to the pleural form: hernias

Line 242. Please add an article before the word inguinal: an inguinal

Line 245. Nine should be spelled out

Line 268. Please add a comma after the word trauma

Line 269. Please add a comma after the word functioning

Line. 273. Please add a comma after the word prick

Comment 2) The number of cases included is low and this can create a statistical problem. Please discuss it.

Comment 3) The statistical analysis is missing in this study. For pre-review paper, there must be a statistical analysis. In case there is not so much data or the number of replications is low, the power statistic of the trial should be explained.

Comment 4) The clinical and ultrasound examination were done before, immediately after and three months after surgery. Could you please discuss the reason that you choose three months after surgery as the third examination time, and why no long-term follow-up was done!

Comment 5) Line 58-60: The contents of the hernial sac may include, among others: omentum, fat, uterus, small intestine, colon, bladder and spleen. Is there any article that reports spleen as an inguinal hernia content in dogs? If yes, please write the references otherwise revise it; however, the sentence is unclear and should be revised.

Comment 6) Please talk about the study limitations.

Comment 7) Please discuss the complication rate of the other open surgical and laparoscopic methods in small animals and compare these results with your results and assess the outcomes. Is there any significant difference between these methods? (you reported the results but did not compare it with the control group (for example open surgery vs. laparoscopic method) or results reported by other studies)

Comment 8) The discussion is, unfortunately, week and without a strong argument.

Comment 9) Line 206-207: The reference number 17 is not explaining the written part.

Comment 10) The authors did not talk about the advantages and disadvantages of the open and laparoscopic methods.

Comment 11) Line 223-224: The written part is not explaining the mentioned reference correctly (Ref. no. 19).

Best regards

Reviewer #4: The manuscript entitled “Percutaneous Internal Ring Suturing – minimally invasive technique for inguinal hernia repair in dogs” is a first description in dogs (n = 9) of an already described surgical technique for inguinal herniorrhaphy in humans (Percutaneous Internal Ring Suturing). The subject is important for companion animal surgery, due to absence of this the theme in the recent literature and the minimal necessity of expertise in endoscopic surgery and also minimal complications observed during the postoperative period. Additionally, the authors described perfectly clear the surgical technique and images are illustrative and have good quality.

However, the discussion about the advantages of the technique is weak, lacking comparison with previous literature about surgical time, postoperative pain and even costs. Also, the conclusions are poorly established in consequence of weak discussion and small number of cases.

Although there are no groups to be compared, the means are shown with standard deviation, but no normality tests of the data were mentioned by the authors.

The authors didn't expose the limitations of their study and technique, probably the size of the dog and weight should be part of discussion and limitations as well.

This is a good quality material and could be submitted as short communication elsewhere or even in Plos One as "Registered Report Protocol" if the authors intend to do future study comparing the different techniques.

Reviewer #5: Abstract in this form is need to be rearranged, please divide it into background, aim, Methods, results and conclusions.

Line 40: do you mean “through” instead of trough.

line 41: the line need to rephrased because it's confusing"including, 18-gauge injection needle with non absorbable 2-0 poly filament"

line 42: please explain that more than one dog was suffering from bilateral inguinal hernia therefore operation done on 9 dogs

abstract stated no conclusions please add it

introduction

line 54: Waters etal 1993 referred to 16 dogs of 22 dogs 73% older than 2 years were female, please rephrase.

line 61-67: please describe other complications of open surgery approach that make laparoscopic approach more safer.

line 69:what reports please insert citation and clarify if these reports "experimental study or clinical cases" and if later applied,please describe what make this research unique.

line 74-77: it need to be rearranged to come before paragraph starting at line 69.

line 78: This part need to be described more extensively as this is the first report of PIRS in dogs and this is very interesting aspect.

line 82- 91: there are a lot of redundancy please rewrite.

line 83:This is the only laparoscopic................etc to tissue injury, please rephrase or add till now.

please add clear aim at the end of introduction part.

Material&Methods

please add inclusion criteria for this research

line 107-109: belong to results section.

line 111: is confusing , do you mean (table1) or Is there missing word after on?

line 114: what's current blood test? please, specify and omit "there were no contraindications to general anesthesia"

line 122: please add an approximate time of operation including anesthesia and Pneumoperitoneum.

line 131-132: no need to re-emphasize that procedure was done under anesthesia.

line 133: please add in bracket gas that used (co2)

please add in bracket brand of insufflation device.

line 141: please specify in details site of needle insertion.

line 158: the subsection need to be rearranged before anesthesia.

The operative technique is extensively described as expected and it's one of this paper strength.

Results:

line 168: What do you mean with "without necessity to conversion"

line 171: Is this time including induction of Pneumoperitoneum, if so please,add in a bracket.

please rearrange ultrasound findings to be after first paragraph.

line 181: please add approximate percentage of possible complications.

Figures are excellent and captions are descriptive.

Discussion:

line 210: 12G or 18G

in general you should state your findings and compare them with other reports, and describe why this technique is superior to other technique.

for example : this approach requires one port and other need 2-3 ports, reduce expected trauma, complications......etc.

again there is no conclusions at end of discussion.

you need to state "a take home message", do you use of this procedure instead of common surgical technique.

what are expected limitations of this technique?

is it cost effective compared to traditional route?

what are expected complications compared to traditional technique?

Are there contraindications to this technique other than known contraindications for traditional surgical practice ?

6. PLOS authors have the option to publish the peer review history of their article (what does this mean?). If published, this will include your full peer review and any attached files.

Reviewer #1: No

Reviewer #2: No

Reviewer #3: Yes: Masoud Aghapour

Reviewer #4: No

Reviewer #5: No

---

## [Author Response · Author response to Decision Letter 0]

14 May 2020

Dear PLOS ONE Editors, Dear Revivers.

At the beginning I would like to thank you very much for a thorough review of our article under the original title „Percutaneous Internal Ring Suturing - minimally invasive technique for inguinal hernia repair in dogs.” All authors are grateful for the reviewers' recognition of the topic of the article as interesting and useful in minimally invasive surgery in animals.

In a separate file named "Respond to Reviewers" there is a more accurate answer to the suggestions of reviewers.

Kind regards,

Przemysław Prządka

---

## [Decision Letter · Decision Letter 1]

25 Jun 2020

Laparoscopic assisted percutaneous herniorrhapy in dogs using PIRS technique.

PONE-D-19-32639R1

Dear Dr. Prządka,

We’re pleased to inform you that your manuscript has been judged scientifically suitable for publication and will be formally accepted for publication once it meets all outstanding technical requirements.

Kind regards,

L. Miguel Carreira, PhD, MSc,DTO,Ps-Grd,DMD,DVM

Academic Editor

PLOS ONE

Additional Editor Comments (optional):

Reviewers' comments:

Reviewer's Responses to Questions

**Comments to the Author**

1. If the authors have adequately addressed your comments raised in a previous round of review and you feel that this manuscript is now acceptable for publication, you may indicate that here to bypass the “Comments to the Author” section, enter your conflict of interest statement in the “Confidential to Editor” section, and submit your "Accept" recommendation.

Reviewer #2: All comments have been addressed

Reviewer #5: All comments have been addressed

2. Is the manuscript technically sound, and do the data support the conclusions?

Reviewer #2: Yes

Reviewer #5: Yes

3. Has the statistical analysis been performed appropriately and rigorously? 

Reviewer #2: N/A

Reviewer #5: N/A

4. Have the authors made all data underlying the findings in their manuscript fully available?

Reviewer #2: Yes

Reviewer #5: Yes

5. Is the manuscript presented in an intelligible fashion and written in standard English?

Reviewer #2: Yes

Reviewer #5: Yes

6. Review Comments to the Author

Reviewer #2: dear author

thanks for doing all the needed corrections in reviewing. good luck and with best regards

Reviewer #5: the research has a new good idea for inguinal hernia treatment and authors made all addressed comments

7. PLOS authors have the option to publish the peer review history of their article (what does this mean?). If published, this will include your full peer review and any attached files.

Reviewer #2: No

Reviewer #5: No

---

## [Editor Report · Acceptance letter]

1 Jul 2020

PONE-D-19-32639R1 

Laparoscopic assisted percutaneous herniorrhapy in dogs using PIRS technique. 

Dear Dr. Prządka:

I'm pleased to inform you that your manuscript has been deemed suitable for publication in PLOS ONE. Congratulations! Your manuscript is now with our production department. 

Kind regards, 

on behalf of

Dr. L. Miguel Carreira 

Academic Editor

PLOS ONE